# Cellular census of human fibrosis defines functionally distinct stromal cell types and states

Thomas B. Layton[1], Lynn Williams[1], Fiona McCann [1], Mingjun Zhang [2], Marco Fritzsche[1], Huw Colin-York[3], Marisa Cabrita[1], Michael T. H. Ng [3], Marc Feldmann[1], Stephen Samson [1], Dominic Furniss [4], Weilin Xie[2,5] & Jagdeep Nanchahal [1]✉

Fibrotic disorders are some of the most devastating and poorly treated conditions in developed nations, yet effective therapeutics are not identified for many of them. A major barrier for the identification of targets and successful clinical translation is a limited understanding of the human fibrotic microenvironment. Here, we construct a stromal cell atlas of human fibrosis at single cell resolution from patients with Dupuytren's disease, a localized fibrotic condition of the hand. A molecular taxonomy of the fibrotic milieu characterises functionally distinct stromal cell types and states, including a subset of immune regulatory ICAM1+ fibroblasts. In developing fibrosis, myofibroblasts exist along an activation continuum of phenotypically distinct populations. We also show that the tetraspanin CD82 regulates cell cycle progression and can be used as a cell surface marker of myofibroblasts. These findings have important implications for targeting core pathogenic drivers of human fibrosis.

[1] The Kennedy Institute of Rheumatology, Nuffield Department of Orthopaedics, Rheumatology and Musculoskeletal Sciences, University of Oxford, Oxford, UK. [2] Department of Inflammation Research, Celgene Corporation, San Diego, CA, USA. [3] MRC Human Immunology Unit, Weatherall Institute of Molecular Medicine, University of Oxford, Headley Way, Oxford, UK. [4] Nuffield Department of Orthopaedics, Rheumatology, and Musculoskeletal Sciences, University of Oxford, Botnar Research Centre, Oxford, UK. [5] Present address: Institute of Materia Medica, Shandong First Medical University & Shandong, Academy of Medical Sciences, Jinan, Shandong Province, P. R. China. ✉email: jagdeep.nanchahal@kennedy.ox.ac.uk

Fibrosis is defined as the accumulation of excess matrix proteins and contributes to a significant proportion of the mortality in developed nations[1,2]. Important mediators of fibrosis are collagen-producing stromal cells, the activation of which coordinates a pathogenic system that can compromise organ function[1,2]. To develop effective therapeutics, we must first have a detailed understanding of the states, subtypes and functional properties of fibrotic stromal cells[3–7]. A major challenge in studying fibrosis is the availability of well-characterised patient samples to dissect the molecular landscape of these disorders.

Patients with localized fibrotic diseases are a rich source of readily accessible early stage tissue[8]. Dupuytren's disease (DD) is a common and progressive fibroproliferative disorder of the palmar and digital fascia of the hand and, in developed nations affects 12% of those aged 55 years, increasing to 29% of those 75 years and older[9]. The initial clinical presentation is the appearance of a firm nodule in the palm that expands into fibrous collagenous cords extending into the digits. Dupuytren's nodules, which represent the early stage of the disease, are a highly cellular fibrotic ecosystem and are an important model to examine developing fibrosis in humans.

In this study, using clinical samples from patients with DD, we construct a molecular taxonomy of human fibrosis. Our single cell atlas of the fibrotic milieu elucidates functionally distinct stromal cell types and states, including ICAM1[+] fibroblasts and CD82[high] myofibroblasts that contribute discrete pro-fibrotic functions. In addition, we demonstrate that the tetraspanin CD82 is expressed on human myofibroblasts and functions to regulate cell cycle progression in this population.

## Results

**Single-cell profiling of developing human fibrosis.** To profile human fibrosis, we examined Dupuytren's nodules using mass cytometry (CyTOF) and single-cell RNA-seq, constructing a single-cell atlas of an active and cellular fibrotic microenvironment (Fig. 1a–e, Supplementary Fig. 1a–f). We generated a dataset yielding high quality profiles of over 300,000 cells from 18 patients (12 single cell RNA-seq and 6 CyTOF) to study cellular heterogeneity in this pathological ecosystem (Fig. 1a–e, Supplementary Fig. 1a–f). Initially, we employed graph-based clustering[10] that resolved a complex disease landscape comprised of fibroblasts, myofibroblasts, immune, pericytes, cycling and endothelial cells (Fig. 1b–e, Supplementary Fig. 1c, d). Next, we performed a GWAS meta-analysis of DD from UK Biobank and the BSSH-GODD study[11], and mapped the associated regions to candidate genes and individual cellular subtypes using SNPsea[12]. GWAS-associated SNPs were strongly associated with particular stromal subtypes (Supplementary Fig. 2a, b), supporting our classification. Marker genes for pericytes, included *JAG1* and *MCAM*, *SFPR4* and *PLA2G2A* for fibroblasts and myofibroblasts were marked by *MMP14* and *MAFB*. The single-cell RNA-seq and CyTOF aligned with significant markers for stromal cell types, where CyTOF further elucidated cell surface expression of these markers for fibroblasts, myofibroblasts and pericytes.

**A dynamic and immunoregulatory ICAM1[+] fibroblast in fibrosis.** We then undertook detailed characterisation of fibroblasts and myofibroblasts. Paired single-cell profiling using scRNA-seq and CyTOF uncovered distinct molecular profiles of ACTA2[−] fibroblasts (CD9, PLA2G2A, C1R and CXCL14) and ACTA2[+] myofibroblasts (α-SMA, ITG-β1, TGF-β1, β-catenin and MMP14). (Supplementary Fig. 2c–g). Myofibroblasts were the predominant cell type representing ~60% of the total stromal cell population (Supplementary Fig. 2d). This observation is consistent with previous histological descriptions of DD nodules

that showed these structures are composed of densely packed myofibroblasts[13]. Functional annotation[14] of cell marker genes using gene ontology (GO) revealed a diversification of function, with inflammation, proliferation and extracellular matrix (ECM) remodelling pathways localising to distinct cell types (Fig. 2a, Supplementary Fig. 2e). Myofibroblasts showed strongest enrichment for pathways involved with cell contraction and ECM remodelling (Fig. 2a, Supplementary Fig. 2e) and core enrichment genes, including *ACTA2, TPM2* and *POSTN*. In contrast, fibroblasts showed an enrichment for genes involved with 'chemokine activity' and 'immune response' pathways (Fig. 2a, Supplementary Fig. 2e), including *CXCL14, CXCL8, IL6, C1R* and *PLA2G2A*.

Next, we performed a focused analysis of ACTA2[−] fibroblasts, subsampling this population from the entire dataset and repeating the computational workflow used to delineate major cell types. Graph-based clustering of transcriptomic profiles defined three major subsets, CD34[+], PDPN[+] and ICAM1[+] fibroblasts (ICAM1[+]IL6[high], ICAM1[+]IL6[low]) (Fig. 2b–d, Supplementary Fig. 3a, b). Moreover, these populations were conserved at the protein level in the CyTOF dataset. GO analysis of fibroblast marker genes demonstrated ICAM1[+]IL6[high] fibroblasts were enriched for immune responsive pathways driven by the expression of several chemokines including *IL6* and *IL8* (Fig. 2d, e). We noted ICAM1[+]IL6[high] fibroblasts were conserved across multiple patient samples and confirmed this subset showed the highest protein expression of IL-6 and IL-8 (Supplementary Fig. 3b, c) using flow cytometry of freshly isolated DD nodular cells. Subsequently, to explore potential relationships between subsets we applied diffusion maps to the fibroblasts. This uncovered a complex topography with discrete trajectories linking CD34[+] and ICAM1[+] subsets with PDPN[+] fibroblasts, suggesting a putative underlying developmental path (Supplementary Fig. 3d).

Next, we sought to define the dynamics of fibroblast subsets in fibrosis pathogenesis. To assess this, we used flow cytometry to determine their proportions within two distinct Dupuytren's structures, the early disease state myofibroblast and immune cell-rich nodule[15,16] and later disease stage matrix-rich cord[13,17] (Fig. 2f, h, Supplementary Fig. 3e). We observed a higher proportion of ICAM1[+] fibroblasts in nodules, which have been shown to harbour the majority of inflammatory cells in DD and are present at the early stages of the disease (Fig. 2h). Subsequently, we tested whether ICAM1[+] fibroblasts could induce immune cell chemotaxis as predicted by their gene expression profiles. For this, we sorted freshly isolated ICAM1[+] and ICAM1[−] fibroblasts (CD45[−]CD31[−]CD146[−]ITGβ1[low]) from Dupuytren's nodules and incubated each with THP-1 mononuclear immune cells (Fig. 2g). This confirmed that ICAM1[+]IL6[high] fibroblasts produced significantly higher immune cell chemotaxis (Fig. 2g). Together, this identifies a dynamic ICAM1[+]IL6[high] fibroblast in human fibrosis which act to promote immune-cell recruitment.

**Distinct myofibroblast states along an activation continuum.** Myofibroblasts are central mediators of the dysregulated wound-healing programme that defines fibrosis[3,4], therefore we studied this population in detail. Graph based clustering of the single cell RNA-seq data defined four major subsets (Fig. 3a) that included a cycling population (Cycling MFB) (Fig. 3a, b). After this, we sought to confirm that proliferating stromal cells (Ki67[+]) were myofibroblasts (Fig. 3c–e, Supplementary Fig. 4a–d). Using flow cytometry, we stained freshly disaggregated nodular cells and gated on myofibroblasts (CD45[−]CD31[−]CD146[−]ITGβ1[high]) and fibroblasts (CD45[−]CD31[−]CD146[−]ITGβ1[low]) and demonstrated Ki67[high] cells were a subset of myofibroblasts. In the single cell

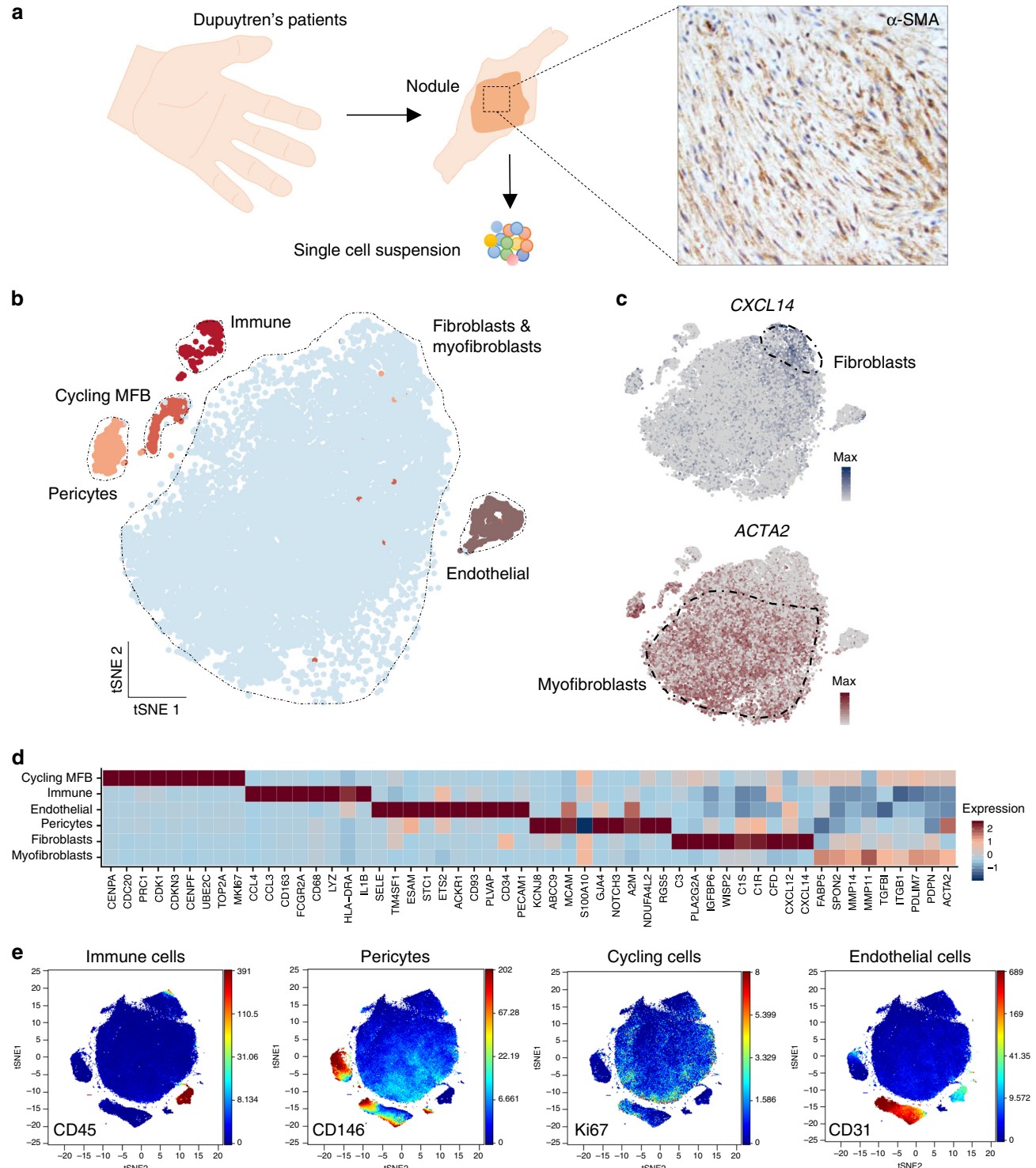

**Fig. 1 Single-cell profiling of developing human fibrosis. a** Schematic illustrating experimental protocol for single-cell profiling of Dupuytren's nodules with representative immunohistochemistry image showing α-SMA protein expression in nodules. **b, c** tSNE projections of single cell RNA-seq for 36,864 cells from the first batch ($n = 6$ DD patients) showing major cell types (**a**) and fibroblasts (*CXCL14* expression) and myofibroblasts (*ACTA2* expression) (**b**) in Dupuytren's nodules. Scale bar in scaled log(UMI + 1). Cycling MFB represents cycling myofibroblasts. **d** Heatmap of single cell RNA-seq showing z-score normalised mean expression of cell type marker genes in nodular cells. ($n = 12$ DD patients). **e** tSNE projections of CyTOF analysis from representative Dupuytren's nodule, coloured by normalised protein expression of cell type markers ($n = 6$ DD patients).

RNA-seq, a second subset was characterised by lower expression of *ACTA2* and intermediate expression of fibroblast marker genes (*CXCL14, PLA2G2A* and *C1R*), we termed the ACTA2low myofibroblast (Fig. 3a, b, Supplementary Fig. 5a). A third comprised a discrete CD82highOX40L+ population (CD82, OX40L, MMP11,

MMP14 and CD82) (Fig. 3a, g, Supplementary Fig. 5b–e). Finally, a fourth less distinct community we termed the intermediate myofibroblast (Fig. 3a, b, Supplementary Fig. 15a). GO analysis demonstrated that CD82highOX40L+ myofibroblast marker genes showed the strongest enrichment for cell contraction pathways

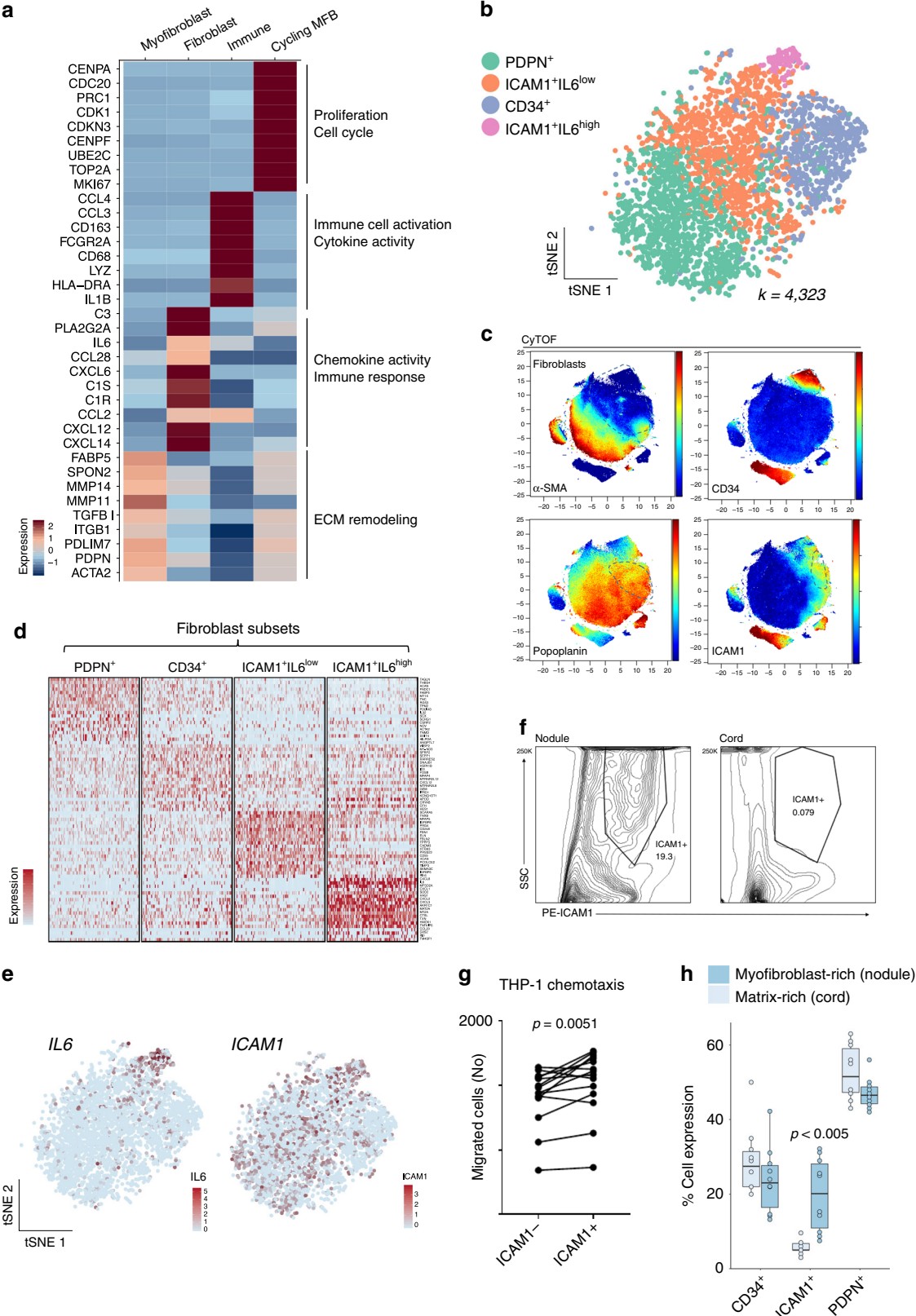

(Fig. 3f). In contrast, differential gene expression and pathway analysis revealed ACTA2^low myofibroblasts resembled a general fibroblast phenotype, without clear characteristics of any one subtype. In addition to *ACTA2*, significant ACTA2^low myofibroblasts markers were general fibroblasts markers including PLA2G2A and CXCL14. ACTA2^low myofibroblasts also shared

pathways enriched in fibroblasts such as 'interleukin signalling' and 'chemotaxis' (Fig. 3f).

As we discovered the membrane tetraspanin *CD82* marked the myofibroblast population showing high expression of *ACTA2* and *ITGB1* (CD82^high OX40L^+ myofibroblast), we confirmed its co-expression with α-SMA and ITGβ-1 proteins (Fig. 3g, h,

**Fig. 2 A dynamic and immunoregulatory ICAM1$^+$ fibroblast in fibrosis. a** Heatmap of single cell RNA-seq showing $z$-score mean expression of genes driving GO signatures in cell types with exemplar GO terms annotated. GO analysis performed on significant cell type markers (two-sided Wilcoxon Rank sum Test, FDR correction, $p$-adjust < 0.01). **b** tSNE projection of fibroblasts coloured by major subsets ($n = 12$ DD patients) in single cell RNA-seq. **c** tSNE projections of CyTOF analysis for representative DD nodule showing three major fibroblasts subsets ($n = 6$ DD patients). Scale bar represents normalised protein expression. **d** Heatmap of single cell RNA-seq showing mean $z$-score expression of top 10 marker genes for each fibroblast subset ($n = 12$ DD patients). **e** tSNE projections of single cell RNA-seq showing fibroblasts coloured by *ICAM1* and *IL6* expression in scaled (log(UMI + 1) ($n =$ DD 12 patients). **f** Representative density plots of flow cytometry analysis showing ICAM1$^+$ fibroblast in freshly isolated cells from matched DD nodule and cord ($n = 8$ DD patients). **g** Dot plot of chemotaxis assay showing migration of THP-1 cells after incubation with ICAM1$^{-/+}$ sorted populations ($n = 11$ DD patients). Two-sided paired $t$ test, $p$ value = 0.0051. **h** Box and whisker plots of flow cytometry analysis showing the percentage of cells (proportion) for fibroblast subsets (CD34$^+$, ICAM1$^+$ and PDPN$^+$) in Dupuytren's nodules and cords as a proportion of total fibroblasts. Two-sided unpaired $t$ test, mean ± SEM ($n = 8$ DD patients). ICAM1 range 3–32%, median 8% and box bounds 5–17%. CD34 range 13–50%, median 25% and box bounds 22–30%. PDPN range 42–63%, median 48% and box bounds 45–55%. Box bounds are first and third quantiles.

Supplementary Fig. 5f, g) using flow cytometry and multiplex immunofluorescence. This validated a cell surface marker of human myofibroblasts and showed tight co-expression of CD82 with established myofibroblast markers α-SMA and ITG-β1 (Supplementary Fig. 5f). Finally, using immunohistochemistry we confirmed CD82$^+$ myofibroblasts were enriched in Dupuytren's nodules as compared to cord (Supplementary Fig. 5g).

The overall topography of the myofibroblast clusters suggested an overarching trajectory structure in which the two distinct cell populations represent diverging ends of a continuum separated by an intermediate cellular state (Fig. 3a). This concept reflects a central principle of myofibroblast biology, which describes an 'activation' paradigm along a differentiation path towards a fully pathogenic phenotype[18,19]. Supporting this concept, principal component analysis (PCA) of the myofibroblasts demonstrated the first principal component to be strongly associated with *ACTA2* expression and inversely correlated with a gene signature of ACTA2$^{low}$ myofibroblasts (Supplementary Fig. 6a, b).

To better understand this activation signature, we used diffusion maps to reconstruct a potential developmental path. In accordance with the PCA, the first two diffusion map components[20,21] captured a trajectory associated with increasing *ACTA2* and *CD82* expression (Supplementary Fig. 6c–f). Fitting a principal curve[22] through the first two diffusion components enabled us to define a dynamic gene module along this continuum (Supplementary Fig. 6c). We observed a coordinated decrease in the ACTA2$^{low}$ myofibroblast markers and progressive increase in CD82$^{high}$OX40L$^+$ myofibroblast markers along this path (Supplementary Fig. 6f). This observation is compatible with the CD82$^{high}$OX40L$^+$ myofibroblast representing a fully 'activated' phenotype, at the terminal stage of a differentiation path. In light of this finding, we performed RNA velocity analysis on human myofibroblasts to explore potential directionality in trajectories using the proportion of spliced and unspliced reads. This demonstrated a complex vector field showing a transition from cycling myofibroblasts into the major cell populations and a putative dynamic equilibrium moving between CD82$^{high}$ and ACTA2$^{low}$ myofibroblasts.

After this, we looked for the spatial expression of CD82$^{high}$ myofibroblast markers in DD nodules using immunohistochemistry. Within nodules, CD82$^{high}$OX40L$^+$ myofibroblast markers (e.g. MMP11, CD82 and MMP14) had the highest expression in densely packed, α-SMA$^+$ myofibroblast-rich foci (Supplementary Fig. 7).

**Functional heterogeneity of myofibroblasts in human fibrosis**. We next investigated whether myofibroblasts along the activation trajectory were phenotypically distinct. We sorted CD82$^{high}$ and CD82$^{low}$ myofibroblasts and performed traction force microscopy (TFM) to measure cell contractility at single-cell resolution (Fig. 4a–d). As force generation is a key in vivo characteristic of

myofibroblasts it is a highly relevant functional readout. In addition, pathway analysis of CD82$^{high}$ myofibroblasts associated this population with cell contraction pathways and thus TFM enabled validation of a functional characteristic predicted in the single-cell RNA-seq (Fig. 4a). This demonstrated human myofibroblasts are functionally heterogeneous with CD82$^{high}$ myofibroblasts marking a highly contractile population (Fig. 4b–d). In addition, CD82$^{high}$ myofibroblasts had distinct biophysical profiles with localised areas of high force magnitude that enables these cells to bind and remodel matrix proteins (Fig. 4b).

To further understand the functional role of *CD82*, we performed bulk RNA-sequencing following siRNA mediated knockdown in freshly isolated myofibroblasts (Fig. 4e, Supplementary Fig. 8a–d). This revealed that *CD82* regulated a wide array of genes and GO analysis of *CD82* regulated genes demonstrated an enrichment for diverse processes, including cell cycle and p53 signalling (Supplementary Fig. 8e). Given this, we sought to validate a functional role of CD82 in myofibroblasts. Using a flow cytometry-based cell cycle assay and siRNA mediated knockdown of *CD82*, we confirmed CD82 directly regulated the myofibroblast cell cycle, promoting entrance into the G2–M phase (Fig. 4e). Together, these results support a role of CD82 in maintaining the proliferative potential of myofibroblasts and may explain how these cells accumulate in this condition.

To explore the relevance of our findings in other fibrotic diseases, next we examined CD82$^+$ stromal cells in a murine model of pulmonary fibrosis (Fig. 4f–h, Supplementary Fig. 9a–c). Remarkably, we discovered that *Cd82* marked a distinct Pdgfra$^+$ fibroblast which was the most upregulated population during fibrosis progression (Fig. 4f–h, Supplementary Fig. 9b, c). Based on these findings, we looked for CD82 protein expression in human pulmonary fibrosis (IPF), co-staining for pertinent markers of Pdgfra$^+$ fibroblasts (PDGFR-α and COL13A1) (Fig. 4i, Supplementary Fig. 9d). This confirmed the presence of CD82$^+$ fibroblasts in human IPF adjacent to fibrotic foci. Collectively, these data show a Cd82$^+$Prgfra$^+$ fibroblast in murine and human pulmonary fibrosis and demonstrate that CD82 marks distinct myofibroblast and fibroblast populations across visceral and localised human fibrosis. It is therefore possible that at different anatomical sites undergoing fibrosis CD82 may drive diverse cellular responses to propagate this disease process.

### Discussion
The fibrotic microenvironment houses multiple cell subpopulations with diverse genetic and phenotypic characteristics[23–26]. How this heterogeneity emerges in developing fibrosis remains unclear and a major barrier to this understanding is the limited availability of high-quality human samples[8,27]. Here, we build a single cell atlas of a human fibrotic microenvironment and describe functionally distinct fibroblast and myofibroblast types and states. A number of recent studies have utilised single cell

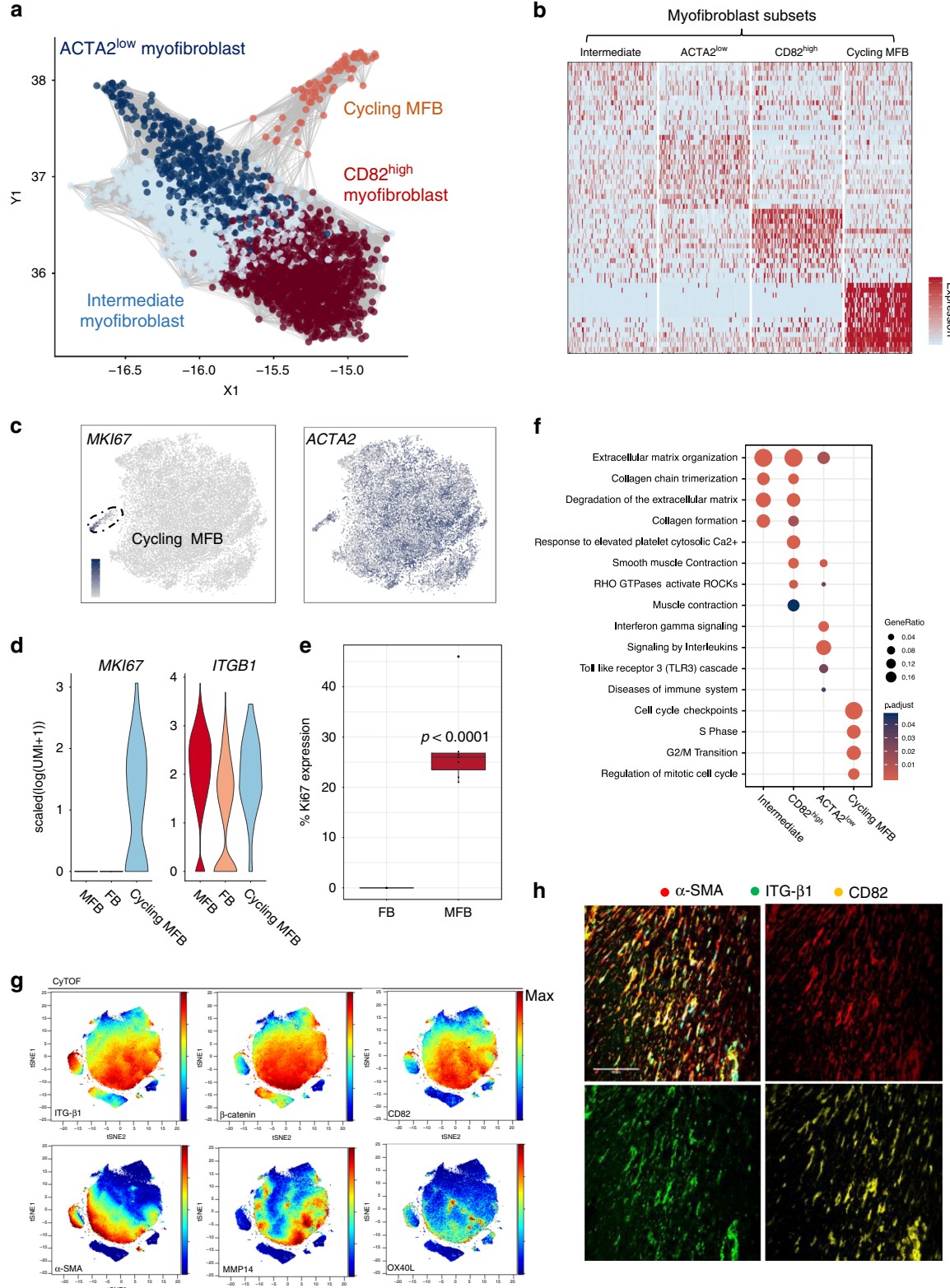

RNA-seq to profile human fibrotic disease, including lung and liver fibrosis, but these reports highlight the challenge in studying small biopsy samples from these disorders[28]. Crucially, our results from Dupuytren's nodules enabled description of stromal cells from a complete fibrotic ecosystem, excised in its entirely, and thus forms a powerful resource for future studies in fibrosis.

Fibroblasts are increasingly recognised as central mediators of diverse pathological systems and here we uncovered three major subsets (ICAM1+, CD34+ and PDPN+) in human fibrosis. Crucially, we discovered a dynamic immune regulatory ICAM1+IL6high fibroblast expressing high levels of chemokines and exhibiting a direct chemotactic activity. In addition, we show this population is expanded in developing and immune-cell rich stages of fibrosis in Dupuytren's nodules. We have previously shown in DD immune cell factors, including TNF, are crucial for myofibroblast development and activation[16]. Thus, together these

**Fig. 3 Distinct myofibroblast states along an activation continuum. a** Force directed graph (Fruchterman–Reingold layout) of single-cell RNA-seq showing myofibroblasts coloured by major subsets ($n = 12$ DD patients). MFB represents myofibroblast. **b** Heatmap of single cell RNA-seq showing the $z$-score expression of top ten myofibroblast subset markers. Two-sided Wilcoxon Rank Sum Test with FDR correction (BH correction ($n = 12$ DD patients). **c** tSNE projections of single cell RNA-seq showing myofibroblasts coloured by the expression of ACTA2 and MKI67 in scaled log(UMI + 1) ($n = 12$ DD patients). **d** Violin plots showing gene expression of MKI67 and ITGB1 in fibroblasts and myofibroblasts in scaled(log(UMI + 1)) from single cell RNA-seq. **e** Box and whisker plot of flow cytometry analysis showing Ki67 protein expression in myofibroblasts ITGβ1[high] myofibroblasts (range 21–46%, mean 28% and box bounds 24–27% representing first to third quantiles) and ITGβ1[low] fibroblasts (range 0–0%, mean 0.0% and percentiles 0%). Two-sided unpaired $t$ test, $p$ value = 0.00014, mean ± SEM. ($n = 8$ DD patients). **f** Dot plot of single cell RNA-seq showing pathways enriched in myofibroblast subsets. Gene ratio is number of marker genes associated with pathway and $p$.adjust is adjusted $p$ value (two-sided Wilicoxon Rank Sum test, BH FDR-correction). **g** tSNE projections of CyTOF analysis for representative DD patient showing distinct CD82[high]OX40L+ myofibroblast. Scale bar is normalised protein expression. ($n = 6$ DD patients). **h** Confocal images of immunofluorescence showing co-expression of CD82, α-SMA and ITG-β1 in DD nodules ($n = 3$ DD patients). Scale bar 20 μm.

findings support ICAM1+IL6[high] fibroblasts as key drivers of inflammation that feedbacks to sustain stromal activation during fibrosis progression.

Tetraspanins are a family of highly conserved proteins containing four transmembrane domains with diverse functions in human health and disease. CD82 represents a canonical membrane tetraspanin, most studied in the context T-cell activation and tumour suppression[29]. Molecular functions of CD82 are broad and include directing intracellular signalling[30], binding to integrins[31,32] and modulating p53[33]. To our knowledge CD82 has not been described in relation to myofibroblasts. Here, we show CD82 is a specific protein and gene marker of human myofibroblasts whose expression tightly correlates with α-SMA and ITG-β1. In addition, we demonstrate it acts to regulate myofibroblast cell cycle progression and provides a mechanism by which these cells may accumulate in DD[34].

Our stromal cell census delineates a unified molecular programme of myofibroblasts in fibrosis and demonstrates these cells exist along an activation trajectory housing ACTA2[low], intermediate and CD82[high]OX40L+ subsets, the latter being highly contractile and housing a cycling population. RNA velocity analysis suggests a complex and dynamic topography of this activation signature containing a proliferative subset that supports a precursors pool. In addition, we show CD82 expression follows the activation trajectory that transitions between ACTA2[low] and CD82[high]OX40L+ myofibroblasts.

A central function of myofibroblasts is the generation of traction force which plays a key role in remodelling the matrix and also modulates the activities of the embedded stromal cells in wound healing and fibrosis[7–9]. Using TFM, we validate functional heterogeneity along the myofibroblast activation trajectory marked by CD82 and show CD82[high] myofibroblasts have distinct biophysical profiles exerting high traction force on ECM coated hydrogels. It is reasonable that force generation in this population contributes to their proliferative and ECM remodelling capacity. However, the biophysical profile of CD82[high] myofibroblasts shows distinct force foci of corresponding size and morphology to fibrillary matrix proteins[35] and supports a predominant role of binding to and remodelling of the matrix.

Given the lack of an animal model for DD we are unable to directly validate our findings in an in vivo model, including potential functional and lineage tracing studies of stromal cell populations. In addition, limitations of murine models of IPF must be acknowledged with regards to their relevance to human disease. Nonetheless, the differential expression of CD82 on fibroblasts in visceral fibrosis and myofibroblasts in localised human fibrosis suggests discrete functions during fibrosis at different anatomical sites. We uncovered a functional role of CD82 in human myofibroblasts in localised fibrosis, but the exact molecular function of CD82+ stromal cells in pulmonary fibrosis remains unknown and represents an exciting avenue for future research.

In summary, our findings provide important insights into mesenchymal subpopulations in human fibrosis, classify a division of labour between fibrotic stromal cells and form a powerful translational resource to help inform development of future treatments.

## Methods

**Patient samples.** After approval by the local ethical review committee (REC 07/H0706/81, University of Oxford), tissue samples were obtained with informed consent from patients with DD. Dupuytren's nodular tissue were obtained from individuals with DD undergoing dermofasciectomy. Given the lack of well-defined controls in DD we instead focused on dissection heterogeneity within the fibrotic microenvironment.

**Cell culture.** Cells from DD were isolated from α-SMA-rich nodules as described previously[13]. Tissue samples were dissected into small pieces and digested in DMEM (Lonza) with Type I collagenase (Worthington Biochemical Corporation) + DNase I (Roche Diagnostics) for up to 2 h at 37 °C. Cells were cultured in DMEM with 5% (vol/vol) FBS and 1% penicillin–streptomycin at 37 °C in a humidified incubator with 5% (vol/vol) $CO_2$. Cells before passage 2 were used for experiments and only freshly isolated cells were used for single cell RNA-seq, CyTOF and flow cytometry.

**Antibodies.** Antibodies used in flow cytometry: PE-ICAM1 (Biolegend HA58), PE/Cy7-CD34 (Biolegend 581−BL), APC-PDPN (Biolegend NC-08), APC-CD31 (Biolegend WM59), APC-CD146 (Biolegend SHM-57), FITC-CD146 (Abcam PIHI2), BV421-CD45 (Biolegend HI30), BV605-ki67 (Biolegend Ki67), APC-IL8 (eBioscience BCH), APC-IL6 (Biolegend MQ2-13A5), AF700-α-SMA (R&D Systems ICI420N), APC-CD29 (Biolegend 152/16) and PE-CD82 (Biolegend ASL-24). Antibodies used for immunohistochemistry and Immunofluorescence: α-SMA (Abcam A5228), PLA2G2A (Novus Bio. 620501), CXCL14 (Abcam ab36622), C1R (Sigma-Aldrich P00736), ITG-β1 (Abcam EP1041Y), β-Catenin (Abcam ab16051), MMP14 (Abcam EP1264Y), FAPB5 (Abcam ab84028), IL-11RA (Abcam ab109697), MMP11 (Abcam ab53143) and CD82 (Abcam TS82b)

**Single-cell Isolation.** Following surgical resection DD samples were rapidly transported to the research facility. On arrival, samples were then rinsed in PBS and the nodule isolated using a scalpel. Each nodule was minced using a scalpel and transferred to 5 ml of digestion medium described above. Dissociated cells were then washed in 5% FBS DMEM (Gibco) and passed through a 100 μm cell strainer. This single-cell solution was slowly frozen using a Mr. Frosty Container (ThermoFisher). Single-cell suspensions were thawed on the day of sequencing, diluted at a concentration of 1000 cells/μL in 0.04% BSA/PBS for loading into 10× Chromium Single Cell A Chips.

**Droplet-based scRNA-seq.** Single cell libraries were prepared using the Chromium 3′ v2 platform (10× Genomics, Pleasanton, CA) following the manufacturer's protocol. In brief, single cells were encapsulated into gel beads in emulsions (GEMs) in the GemCode instrument followed by cell lysis and barcoded reverse transcription of RNA, amplification, shearing and 3′ adaptor and sample index attachment. Approximately, 10,000 single cells were loaded on each channel and approximately 1500–6000 cells were recovered. Libraries were sequenced on the Illumina HiSeq 4000 (Paired end reads: Read 1, 26 bp, Read 2, 98 bp).

**Computational analysis.** Sample de-multiplexing, alignment to the GRCh38 human transcriptome and UMI-collapsing were performed using the Cellranger toolkit (v2, 10× Genomics). Aspects of the downstream analysis were done in Seurat R package (Sajita Lab). We first performed quality control on each patient

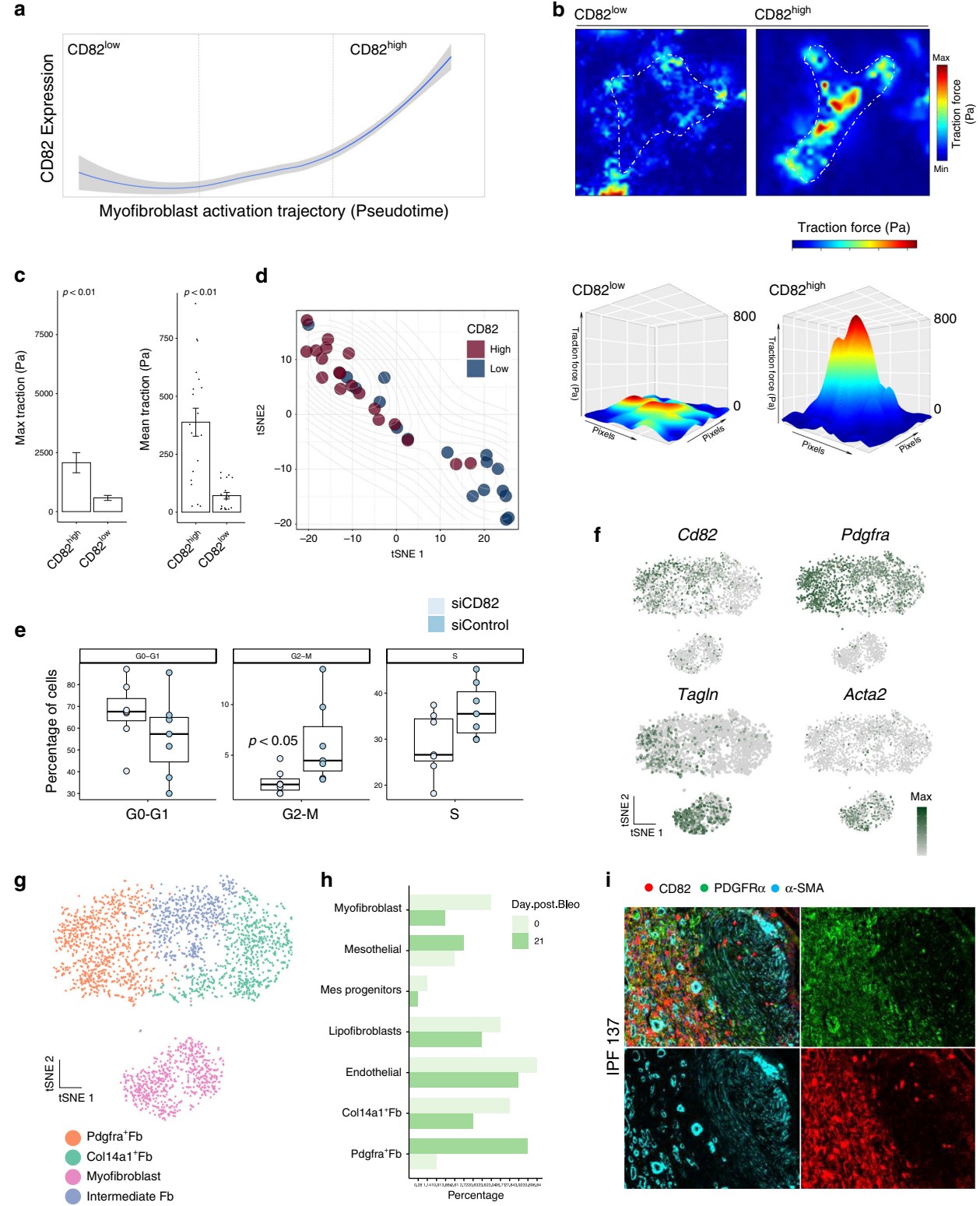

sample separately. We excluded poor quality cells that expressed fewer than 200 genes and with over 10% of UMIs mapping to mitochondrial genes as well as cells that expressed over 4500 genes. After this, the donors were aligned from the first and second batches. Batch correction was performed using Combat as implemented in the 'sva' R package using the default parametric adjustment mode[36]. Batch correction was assessed before downstream analysis and we observed minimal batch effects following the Combat method. We employed a global-scaling normalisation procedure as per[37] calling Seurat's LogNormalize() function. Briefly, the cell ranger UMI count value for each gene was divided by the sum of the total UMI counts per cell to normalise for differences in library size and then multiplied

by a scaling factor that represented the median library complexity (10,000) producing TPM-like values. We then took the log transform of this procedure for downstream analysis. Downstream analysis was performed using all patient donors, and first and second patient batches were displayed separately for visualisation only.

Feature selection was first undertaken by defining highly variable genes using the FindVariableGenes() function (1856 genes). Values were then centred and scaled before input to PCA, which was implemented using the R function 'prcomp' from the 'stats' R package. After PCA, significant PCs were identified using the permutation test implemented using the 'permutation PA' function from the

**Fig. 4 Functional heterogeneity of myofibroblasts in human fibrosis. a** Line plot demonstrating *CD82* expression in myofibroblasts (*loess* smoothed normalised counts ± SE) along pseudotime (myofibroblast activation trajectory) in single-cell RNA-seq (*n* = 12 DD patients). **b** Above, Heatmaps of traction force microscopy analysis showing magnitude of traction force (Pascals) in freshly isolated FACS sorted myofibroblasts (CD82high and CD82low). Below, surface plots of traction force microscopy showing representative force foci from sorted myofibroblasts (Scaled to 800 Pa). Colour scale represents force range per cell. **c** Bar plots of traction force microscopy analysis showing mean and maximum traction force per cell in CD82high and CD82low sorted myofibroblast populations (*n* = 3 DD patients, >15 cells per condition). Two-sided unpaired *t* test, mean ± SEM. *p* Values = 0.0056 (max traction), and 0.0098 (mean traction). **d** tSNE projection of traction force microscopy analysis showing clustering of single cell force signatures in sorted populations (CD82high and CD82low) (*n* = 3 DD patients, >15 cells per condition). **e** Box and whisker plot of flow cytometry analysis showing cell cycle phases in siControl and siCD82 transfected DD myofibroblasts (*n* = 3 DD patients). Two-sided paired *t* test, *p* value = 0.023, mean ± SEM. (siRNA range 2.6–85.5%, mean 32.8% and box bounds 9.75–45.2% represent first and third quantiles. siCD82 range 1.1–87.1%, mean 32.7% and box bounds 3.1–59.8% represent first and third quantiles.) **f** tSNE projections of single-cell RNA-seq showing fibroblasts and myofibroblasts in bleomycin murine model marked by the expression of selected genes in scaled log(UMI + 1) (*n* = 2 mice). **g** tSNE projection of single cell RNA-seq showing murine fibroblasts and myofibroblasts in bleomycin model coloured by Louvain clusters (*n* = 2 mice). **h** Bar plot of single cell RNA-seq showing the percentage of stromal cells before (Day 0) and after (Day 21) the installation of bleomycin in murine lung fibrosis model. **i** Confocal images of immunofluorescence showing co-expression of CD82, α-SMA and PDGFR-α in human IPF (*n* = 3 independent IPF patients). Scale bar = 20 μm.

'jackstraw' R package[38]. This test identified 20 significant PCs and these were used as the input to further analysis.

To partition the data into clusters of transcriptionally related cells, we used unsupervised clustering based on the Louvain algorithm with the Jaccard correction implemented in the FindClusters() function. Initially, we used clustering to delineate the major cell types. Cell clusters were then annotated to known biological cell types using canonical marker genes. In many of the patient samples and pooled dataset, we noted a small (<2%) population of proliferative stromal cells. Regression of the cell cycle effect did not significantly influence clustering and therefore we opted not to undertake correction for cell cycle effects. After defining the major cell types we subsampled each (Fibroblasts, myofibroblast, Pericytes and endothelial cells) to identify potentially meaningful sub-clusters. We applied the same steps as described above to delineate sub-clusters including selection of highly variable genes, PCA and selection of significant PCs. As the average UMI count varied between distinct cell types when analysing individual cell types, we performed a second round of quality control to remove doubles and contaminants. We used a set of canonical markers for each cell type and excluded those cells that shared markers of multiple cell types and those cells with a UMI count three standard deviations above the median cell-type UMI count. For visualisation we used the Barnes–Hut approximate version of t-distributed stochastic neighbour embedding (tSNE) using the 'Rtsne' R package.

On inspection of the the PCA projection of the myofibroblast dataset we noted a strong association between *ACTA2 and CD82*+ marker expression to the first principal component. This was apparent in both pooled datasets and in individual patient samples. To investigate this further, we extracted those genes with the top PC1 loading and again saw a strong negative correlation to ACTA2low myofibroblast marker genes (e.g. *PLA2G2A* and *C1R*) and strong positive correlation to *ACTA2* expression and markers of the CD82+ population. Given this, we reasoned that this may represent the transcriptional profile of a putative differentiation or activation pathway. To visualise this, we defined a gene module of the ACTA2low myofibroblasts and CD82+ myofibroblast taking top 100 genes correlated (Spearman) to *ACTA2* and *CXCL14*. Then, we plotted PC1 score of each cell against the average expression of these gene modules minus a control gene set and obtained the Pearson correlation coefficient. To account for differences in library complexity between cells we calculated a control gene set from these scores[39]. This was selected by binning all genes in the dataset and randomly selected 100 genes from each bin that contained every gene of the test gene set. Thus, the control gene set has a comparable distribution of expression levels to that of the ACTA2 and CXCL14 gene-sets and the control gene set is 100-fold larger, such that its average expression is analogous to averaging over 100 randomly-selected gene-sets of the same size as the ACTA2 and CXCL14 gene-sets. This revealed a strong positive correlation to the ACTA2 gene module (*R* = 0.46) and a strong negative correlation to the fibroblast-like gene module (*R* = −0.71) along the first principal component, suggesting a strong transcriptome signature of this putative trajectory in the data.

To assign myofibroblast along the putative activation trajectory we first performed dimensional reduction using diffusion maps, implemented in the R package 'destiny'[40]. The distance metric was the Euclidean distance between pairs of cells in the reduced dimension space of the significant PCs (*n* = 20), as defined above. In concordance with the PCA, the first two diffusion components were highly associated with *ACTA2* expression. We then fitted a principal curve (R package princurve, smoother = 'lowess', *f* = 1/3) through the first two diffusion co-ordinates[41]. As the *λ* value of the curve reflects the arc-length from the beginning of the curve for each point we utilised this to assign each cell to a 'Pseudotime' trajectory. Importantly, this modelled trajectory was focused on capturing myofibroblast differentiation associated with *ACTA2* expression, but did not ascribed any distinct cell type as the myofibroblast precursor. In addition to this approach, we also the 'slingshot' R package to infer trajectories and align cells along developmental pseudo-time. Slingshot revealed a very similar trajectory structure

with a path between ACTA2low and CD82high myofibroblasts. Expression smoothing was performed using generalised additive models implement in the 'gam' function in the 'mgcv' R package. RNA velocity analysis was performed using Velocyto using default parameters.

GO enrichment of cluster markers and differentially expressed genes was performed using the R package 'clusterProfiler'[42] with a Benjamini–Hochberg multiple testing adjustment and a false-discovery rate cut-off of 0.1, using all expressed genes expressed in >3 cells as background. Visualisation was performed using the R packages 'ggplot2' and 'igraph'.

The collection of DNA samples, genotype calling, quality control and imputation of the BSSH-GODD cohort has previously been described[11]. Similarly, information pertaining to the UKBiobank genotype and phenotype data has been previously described[43]. Additional QC thresholds were implemented to the UKBiobank genotyping data to remove potential spurious variants and samples, namely (1) call rate <98%, (2) heterozygosity >3 standard deviation from the mean, (3) individuals with aneuiploidy or demonstrating a discrepancy between self-reported sex and genetically inferred sex and (4) non white British, (5) SNP does not conform the Hardy–Weinberg equilibrium (*p* < 0.004) and (6) MAF < 1%. The UKBiobank imputed data is a combination of the Haplotype Reference consortium and the UK 10 K/1 K Genomes Phase 3 panel comprising 92,693,895 variants and 487,442 individuals, which was reduced to 401,667 after QC. A total of 3736 Dupuytren disease patients were identified within the UKBiobank using a combination of diagnostic codes; ICD-10 (M720, M7204), OPCS (T521, T522, T525, T526, T561 and T562), non-cancer self-reported illness code 20002 (1544), and self-reported operation code 20004 (1535), and were age, sex and genotyping platform matched at a ratio of 1–5 to select 18,680 controls. The summary association statistics of both datasets were calculated using BOLT-LMM[44], and variants with an info score <0.9, a standard error >5 or an estimated minor allele count in cases <3 were removed prior meta-analysis, which was performed using the inverse variance weighted method provided in GWAMA[45].

To generate a normalised count from the single cell RNA expression data, raw counts of each gene of the same cell type were amalgamated and subsequently normalised using DESeq2[46]. Associations between the normalised RNA expression and DD associative variants were identified using SNPsea[12].

**Flow cytometry**. After tissue disaggregation into a single cell suspension, cells were first stained with a panel of fluorescently labelled antibodies to surface antigens, washed with FACS wash buffer (1% bovine serum albumin (BSA), 0.01% NaN₃ in PBS) then fixed using CytoFix (eBiosciences Foxp3 staining buffer set # 00-5523-00) for a minimum of 30 min at 4 °C. After three washes in perm wash (eBioscience Foxp3 staining buffer), intracellular antigens were stained with another panel of fluorescently labelled antibodies, and cells were then washed with perm wash and analysed by flow cytometry (BD LSR Fortessa X20) and FlowJo software. BD comp beads (anti-mouse Ig k #552843) were used to establish compensation settings. Dead cells were deselected using live/dead stain, added to surface staining panel prior to cell fixation (Life Technologies Live/Dead™ Near IR fixable dead cell stain kit #L10119). Isotype controls were used during antibody optimisation.

**FACS sorting**. Dupuytren's myofibroblasts were stained without prior magnetic bead enrichment into CD82high and CD82low based on the top and bottom 30% expressing cells, respectively. Samples were filtered through a 70 μm strainer before sorting commenced. Single-cell sorting was performed using a FACSAria (BD Biosciences). After doublet exclusion, isolated single cells were sorted into 5 ml FACS tubes containing 1 ml of complete medium (5% DMEM + 1% pen/strep). Cell purity was assessed for each sorted population and was approximately 95%. Sorted stromal cell populations were seeded onto hydrogels for TFM.

**Mass cytometry**. Where possible antibodies were purchased from Fluidigm, (San Francisco, CA) or were labelled using metal tags using Maxpar antibody labelling kits (Fluidigm). All antibodies were titrated and used at a concentration between (0.25 and 0.5 μg/ml). For each sample, 1–3 million cells were first stained with a solution containing rhodium DNA intercalator (Fluidigm) to distinguish live/dead, prior to Fc receptor blocking (Miltenyi Biotec). Samples were then stained with a mixture of metal conjugated antibodies recognising cell surface antigens (see staining panel). After washing in Maxpar cell staining buffer (Fluidigm), samples were fixed and permeablised using the FoxP3 transcription factor staining buffer kit (Thermofisher) prior to washing and incubation with metal conjugated antibodies recognising intracellular antigens. Samples were washed twice in cell staining buffer, fixed by incubation with 1.6% PFA (Pierce) for 10 min and finally incubated overnight with iridium DNA intercalator in Maxpar fix and perm buffer (Fluidigm). Prior to acquisition samples were washed twice in Maxpar cell staining buffer and twice in Maxpar water and filtered through a 40 μm cell strainer before being acquired on a Helios mass cytometer (Fluidigm).

After acquisition, all.fcs files in the experiment were normalised using tools within the Helios software and then uploaded to Cytobank (www.cytobank.org) for all gating and further analysis including using the clustering and dimensionality reduction algorithm viSNE.

**CyTOF antibodies**. CD252 169-Tm (3166007B, ML5, Fluidigm), CD34 166-Er (Catalog:3166012B, Clone:581, Fluidigm), CD45 141-Pr (Catalog: 3141009B, Clone: HI30, Fluidigm), HLAD A, B C 142-Nd (Catalog: 3142007B, Clone: HCD57, Fluidigm), CD19 143-Nd (Catalog: 3144007A, Clone:NP6G4, Fluidigm), HLA-DR 89-Y (Catalog: 3173005B, L243, Fluidigm), CD3 189-Y (Catalog: 3158021A, Clone: 24E10, Fluidigm), B-catenin 176-Lu (Catalog: 3147005A, Clone:D10A8, Fluidigm), CD55 174-Yb (Catalogue; 3148015B, Clone: JS11, Fluidigm), CD146 155-Nd (3155006B, P1H12, Fluidigm), CD29 156-Gd (3156007B, TS2/16, Fluidigm), CD82 158Gd (3158025B, ASL-24, Fluidigm), CD90 172-Yb (3173011B, 5E10, Fluidigm), TGFb 150-Nd (3163010B, CloneTW46H10, Fluidigm) CD163 145-Nd (3145010B, GHI/61, Fluidigm), Ki-67 168-Er (3168007B, B56, Fluidigm), CD54 170-Er (3170014B, HA58, Fluidigm), CD68 171-Yb (3171011B, Y1/82A, Fluidigm), CD9 172-Yb (Cat. No. 312102, Biolegend), CD252 150-Nd (326302, Biolegend), PGLA2 172-Yb (MAB5374, R&D Systems), a-SMA 156-Gr (MAB1420, 1A4, R&D Systems), gp38 155-Gd (AF3670-SP, R&D Systems), PDGFRB 154-Sm (SAB4700458, 18A2, Sigma Aldrich), MMP14 149-Sm (MAB9181-SP, 128527 R&D Systems), Cad-11 148-Nd (AF1790-SP, R&D Systems) and FAP 147-Sm (MAB3715-SP, 427819 RD Systems).

**Immunohistochemistry**. Dupuytren's tissue samples were fixed with 4% paraformaldehyde in PBS for 20 min, longitudinally bisected, and embedded in paraffin wax, and 7-μm sections from the cut surface were processed for immunohistochemistry[13]. Sequential sections were stained with mouse monoclonal antibodies. Antibodies were detected using a two-staged polymer enhancer system (Sigma). Murine or Rabbit IgG isotypes at the same protein concentration as the monoclonal antibody solution were used as a control. Images were acquired with an Olympus BX51 Microscope (Olympus).

**Immunofluorescence and confocal microscopy**. Dupuytren's tissue samples were fixed with 4% paraformaldehyde in PBS for 20 min, longitudinally bisected and embedded in paraffin wax, and 7-μm sections from the cut surface were processed for immunofluorescence[13]. Then, the tissue sections were stained with antibodies listed above followed by incubation with fluorescent-dye-conjugated secondary antibodies (Life Technologies). Nuclei were counterstained with DAPI (4, 6-diamidino-2-phenylindole; Sigma-Aldrich) and mounted using Prolong™ Gold antifade (Life Technologies). Fluorescent images were obtained with a confocal system (Zeiss LSM 710).

**Traction force microscopy**. Polyacrylamide (PAA) hydrogels for TFM were prepared as previously described[47]. Briefly, acid washed glass coverslips (18 mm) were incubated with poly-L-lysine (10 μg/ml in H₂0) for 30 mins at 4 °C, then coated with 0.04 μm carboxylate-modified red FluoSpheres (1:5000 in ddH₂0). PAA gel formation was initiated with APS (10% solution in ddH₂0, Sigma) and *TEMED (Sigma)*. Polymerised PAA gels were functionalized with sulfo-SANPAH (Invitrogen) and coated with Type 1 Collagen (200 μg/ml Rat Tail, Thermo Fisher). The Young's modulus of the PAA gels were 2.55 ± 0.5 kPa. Cells were allowed to adhere to the gel for 4–6 h before image acquisition. Images were captured using a ×40 Plan Apo objective on a Zeiss LSM 710 Confocal microscope and the experiments were performed at 37 °C and 5% CO₂ in Phenol red free DMEM (Thermo Fisher) in a microscope stage incubation chamber. Bead positions were acquired before and after the addition of trypsin which removed cells from the gel surface. Bead displacement was tracked with an ImageJ PIV plugin[48] and cellular forces reconstructed using a FTTC algorithm also implemented in ImageJ[48,49].

Downstream analysis and visualisation was performed using MATLAB (Mathworks) and R (R Version 3.5). Single cell force maps of CD82^high and CD82^low sorted cells were first normalised per cell area and scaled before PCA implemented in the "prcomp" function. Significant components were determined as described above for the single-cell expression data using visualisation of a scree plot and/or the Jackstraw procedure. Significant PCs were used as input for tSNE clustering using the Barnes–Hut implementation in Rtsne() function.

**Transfections**. For transfection experiments, cells were seeded at $2 \times 10^5$/6-well plate and the following day transfected with siRNA against CD82 (AM16708) or non-targeting control oligo at 1–20 nM (Life Technologies). All transfections were performed with Dharmafect 1 (GE Healthcare) and Optimem™ (Life Technologies) according to the manufacturer's instructions. Media was replaced 2 h later with 5% FBS phenol red free DMEM (Gibco). Experiments were performed 6 days post transfection.

**Chemotaxis assay**. Totally, $5 \times 10^5$ THP-1 cells (ATCC) were allowed to adhere for 2 h to a Corning Transwell® plate with 8 μM pore in 1% FBS DMEM, the reservoir plate was then replaced with conditioned media generated from freshly sorted DD fibroblasts. For the last 3 days of the assay, the media was replaced with 1% FBS, DMEM. Supernatants harvested, filtered and used neat, alongside CCL2 (2 ng/ml) (Peprotech, London, UK) as a positive control. Cells were allowed to migrate for 6 h, the number of migrated cells were counted in the reservoir plate following the addition of Hoechst 3342 Fluorescent Stain (Life Technologies) at 0.1 μg/ml for 1 h using the Celigo Imaging Cytometer, Nexcelom Bioscience (Lawrence, MA).

**Bulk RNA-seq library preparation**. Passage 2 DD myofibroblasts from 7 donors were transfected with siRNA against *CD82* (AM16708) or non-targeting control oligo at 1–20 nM (Life Technologies). RNA was extracted using Direct-Zol™ RNA kit. The library preparation was started with the NEBNext® Poly(A) mRNA Magnetic Isolation Module followed by the NEBNext® Ultra™ Directional RNA Library Prep Kit for Illumina with the NEBNext® Multiplex Oligos for Illumina® (Index Primers Set 1) and (Index Primers Set 2). The concentration of each library was determined using the NEBNext® Library Quant Kit for Illumina® and High Sensitivity D1000 Screentape Bioanalyzer (Agilent).

**Bulk RNA-seq data analysis**. FASTQ files were assessed using FASTQC followed by the generation of TPM values with kallisto v0.42.4[50]. TPM values were summed to obtain gene-level expression values using tximport and differential expression analysis was undertaken with DeSEQ2[46]. GO enrichment of differentially expressed genes was performed using the R package 'clusterProfiler'[42] with a Benjamini–Hochberg multiple testing adjustment and a false-discovery rate cut-off of 0.1. Visualisation was performed using the R packages 'ggplot2' and 'igraph'.

## Data availability

RNA-Seq data are deposited in Sequence Read Archive (SRA) under primary accession codes PRJNA607098 and PRJNA623191. The data supporting the findings of this study are available within the paper and its Supplementary Information files or on reasonable request from the corresponding author. The source data underlying all figures are provided as a Source Data file.

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

## Acknowledgements

We are grateful to our clinical collaborators Christopher Bainbridge, David Warwick, Lorraine Harry and Dominique Davidson for provision of tissue samples from their patients. We thank the Wellcome Trust Centre for Human Genetics for preparation of the single cell libraries. In addition, we are grateful to the Imaging Centre at the Kennedy Institute of Rheumatology for their support in microscopy.

## Author contributions

J.N., S.S., W.X., M. Feldmann and T.L. conceived and designed the study. T.L., L.W., M.N., D.F., F.M., M.C., M.Z., C.Y., M. Fritsche and H.C.Y. performed and contributed to the analysis of the experiments. J.N. and T.L. wrote the paper with contribution from all co-authors.

## Competing interests

J.N. and M. Feldmann declare that they are co-founders and have equity in 180 Therapeutics, which has exclusively licensed IP from the University of Oxford for the treatment of Dupuytren's disease. The remaining authors declare no competing interests.
