## [Peer Review File · Nature Communications]

Reviewers' comments:

Reviewer #2 (Remarks to the Author):

In general, few new experiments have been added. The study remains largely descriptive. The authors did not follow the suggestions of all reviewers to demonstrate functional relevance of the CD82 positive cells in murine models. Although I agree that there may no models for DD and that the models for skin fibrosis and IPF are not perfect, they do provide interesting mechanistic insights and may thus help to demonstrate a functionally relevant role. The in vitro data that have been added to the ms are alone not convincing. The upregulation of certain chemokines alone does not support the conclusion that this population is functionally relevant. This would be only meaningful together with data that show enhanced recruitment of inflammatory cells in vivo. Moreover, I still believe in the importance of studies in normal wound healing to see whether this population is fibrosis-specific or although occurs in physiological tissue remodeling. I am although still not convinced by the stainings for CD82; in contrast to the images in the previous version with a low number of positive cells, now virtually every PDGFR expressing fibroblast coexpresses CD82. Please show appropriate isotype controls and perform quantification in a significant number of patients to enable final conclusions!

Reviewer #3 (Remarks to the Author):

This report dissects the cellular composition of Dupuytren's disease (DD) fibrotic nodules using single-cell RNA-sequencing and other technologies. Herein the authors find a large number of fibroblast/myofibroblasts and to a lesser extent pericytes, endothelial cells and immune cells. The authors argue they have identified 4 unique fibroblast subsets, including one that expresses chemokines, and two general myofibroblasts subpopulations, one myofibroblast subset expresses high levels of ACTA2 and CD82 and displays increased traction capacity. Therein the authors conclude they have identified novel fibrotic stromal subsets with unique functionality. The authors then try to extend their findings using a murine bleomycin-induced pulmonary fibrosis model, but find that a class of fibroblasts are labeled with CD82 but not ACTA2. This latter finding ends up confusing the message of CD82 as a marker for pro-fibrotic myofibroblasts and leaves the reader unclear as to how the murine model relates to the human findings.

Further, it is unclear if the authors propose that CD82 is primarily involved in proliferation control of stromal cells or something more specific to fibrosis?

It is unclear also if the force traction is associating with cells that are more proliferative or is a sign of a fibrotic phenotype. One way to discern is to show what a proliferating and resting fibroblasts v. proliferating and resting myofibroblasts would score on this traction assay. Do myofibroblast that are have a high proliferative capacity (as is suggested here with the CD82 myofibroblasts) still have a higher traction capacity than highly proliferative fibroblasts. If so, then one could conclude that indeed this assay is assaying an increased traction capacity that relates to myofibroblasts/pro-fibrosis rather than relating simply to increased proliferation capacity, which does required increased traction capacity.

Lastly, is the ACTA2-lo myofibroblast overlapping with any of the 4 fibroblast subsets? The authors in the rebuttal mention an analysis to examine this but this could not be found in the data provided. If so, what was tested in the traction assay may have been fibroblasts-transitioning v. myofibroblasts, which is fine but would be good to nail down.

Overall, the messages need to be simplified and clarified with a rewriting of the manuscript. The messages include: (1) identification of 4 fibroblast subsets in DD disease nodules, one with chemokine expression and therein leukocyte attraction capacity, (2) myofibroblast with traction activity in DD disease express CD82, (3) (It is not clear if the Bleo story fits with these findings).

Further, the figure legends should include more experimental information, for example, it was unclear from the legend where the cells came from in Fig. 4A.

Reviewers' comments:

Reviewer #2 (Remarks to the Author):

In general, few new experiments have been added. The study remains largely descriptive. The authors did not follow the suggestions of all reviewers to demonstrate functional relevance of the CD82 positive cells in murine models. Although I agree that there may no models for DD and that the models for skin fibrosis and IPF are not perfect, they do provide interesting mechanistic insights and may thus help to demonstrate a functionally relevant role. The *in vitro* data that have been added to the ms are alone not convincing. The upregulation of certain chemokines alone does not support the conclusion that this population is functionally relevant. This would be only meaningful together with data that show enhanced recruitment of inflammatory cells *in vivo*. Moreover, I still believe in the importance of studies in normal wound healing to see whether this population is fibrosis-specific or although occurs in physiological tissue remodelling. I am although still not convinced by the staining for CD82; in contrast to the images in the previous version with a low number of positive cells, now virtually every PDGFR expressing fibroblast coexpresses CD82. Please show appropriate isotype controls and perform quantification in a significant number of patients to enable final conclusions!

Many thanks for your comment. In response, we have added a point to the discussion stating that given the lack of an animal model for DD we are unable to directly validate our findings in an *in vivo* model, including potential functional and lineage tracing studies of stromal cell populations. In addition, limitations of murine models of IPF have been acknowledged with regards to their relevance to human disease

We have also addressed comments regarding staining and have added the appropriate isotype controls to Sup Fig 10, in addition to performing quantification of the Col13a1+CD82+ fibroblasts in a significant number of patients as suggested.

Reviewer #3 (Remarks to the Author):

This report dissects the cellular composition of Dupuytren's disease (DD) fibrotic nodules using single-cell RNA-sequencing and other technologies. Herein the authors find a large number of fibroblast/myofibroblasts and to a lesser extent pericytes, endothelial cells and immune cells. The authors argue they have identified 4 unique fibroblast subsets, including one that expresses chemokines, and two general myofibroblasts subpopulations, one myofibroblast subset expresses high levels of ACTA2 and CD82 and displays increased traction capacity. Therein the authors

conclude they have identified novel fibrotic stromal subsets with unique functionality. The authors then try to extend their findings using a murine bleomycin-induced pulmonary fibrosis model, but find that a class of fibroblasts are labeled with CD82 but not ACTA2. This latter finding ends up confusing the message of CD82 as a marker for pro-fibrotic myofibroblasts and leaves the reader unclear as to how the murine model relates to the human findings.

Thank you. We have now added clarified our findings relating to CD82 staining in pulmonary fibrosis in our discussion by stating that differential expression of CD82 on fibroblasts in visceral fibrosis and myofibroblasts in localised human fibrosis suggests potentially distinct functions during fibrosis at different anatomical sites. We have also clarified that whilst we uncovered a functional role of CD82 in human myofibroblasts in localised fibrosis, the exact molecular function of CD82⁺ stromal cells in pulmonary fibrosis remains unknown and represents an exciting avenue for future research.

Further, it is unclear if the authors propose that CD82 is primarily involved in proliferation control of stromal cells or something more specific to fibrosis? It is unclear also if the force traction is associating with cells that are more proliferative or is a sign of a fibrotic phenotype. One way to discern is to show what a proliferating and resting fibroblasts v. proliferating and resting myofibroblasts would score on this traction assay. Do myofibroblast that are have a high proliferative capacity (as is suggested here with the CD82 myofibroblasts) still have a higher traction capacity than highly proliferative fibroblasts. If so, then one could conclude that indeed this assay is assaying an increased traction capacity that relates to myofibroblasts/pro-fibrosis rather than relating simply to increased proliferation capacity, which does required increased traction capacity.

We agree this point requires further clarification. Therefore, we have added a statement to the manuscript discussion regarding the ACTA2^{low}, intermediate and CD82^{high}OX40L⁺ myofibroblasts, the latter being highly contractile and housing a cycling population. A central function of myofibroblasts is the generation of traction force, which plays a key role in remodelling the matrix and also modulates the activities of the embedded stromal cells in wound healing and fibrosis⁷⁻⁹. Using traction force microscopy, we validate functional heterogeneity along the myofibroblast activation trajectory marked by CD82 expression and show that CD82^{high} myofibroblasts have distinct biophysical profiles, exerting high traction force ECM coated hydrogels. The biophysical profile of CD82^{high} myofibroblasts shows distinct force foci of corresponding size and morphology to fibrillary matrix proteins³⁶ and supports a predominant role of binding to and remodelling of the matrix. It is also possible that force generation in this population contributes to their proliferative capacity although we note that only a subset of CD82^{high} myofibroblasts belonged to this rapidly cycling population (Fig 1C & Extended Data A-D). We did not find identify a proliferative compartment in the fibroblast population (Fig 1C, 3C & Extended Data A-D)

Lastly, is the ACTA2-lo myofibroblast overlapping with any of the 4 fibroblast subsets? The authors in the rebuttal mention an analysis to examine this but this could not be found in the data provided. If so, what was tested in the traction assay may have been fibroblasts-transitioning v. myofibroblasts, which is fine but would be good to nail down.

Many thanks for your comment. Using differential gene expression and pathway analysis ACTA2^{low} myofibroblasts more resembled a general fibroblast phenotype, without clear characteristics of any one subtype. In addition to ACTA2, significant ACTA2^{low} myofibroblasts markers (Wilcox test, FDR corrected) were general fibroblasts markers including PLA2G2A and CXCL14, and ACTA2^{low} myofibroblasts shared pathways enriched in fibroblasts such as 'interleukin signalling' and 'chemotaxis', as highlighted in our results section (Fig. 3f).

With regards to the second question, we agree that the fibroblast population is likely to behave similar to the ACTA2^{low} myofibroblasts in the traction force assay. We demonstrate an increase in traction force during myofibroblast differentiation marked by CD82, and as ACTA2^{low} and fibroblasts both share a low expression of CD82 and ACTA2 it is probable these cells both exhibit low contractile properties.

Overall, the messages need to be simplified and clarified with a rewriting of the manuscript. The messages include: (1) identification of 4 fibroblast subsets in DD disease nodules, one with chemokine expression and therein leukocyte attraction capacity, (2) myofibroblast with traction activity in DD disease express CD82, (3) (It is not clear if the Bleo story fits with these findings). Further, the figure legends should include more experimental information, for example, it was unclear from the legend where the cells came from in Fig. 4A.

Many thanks for this very helpful suggestion. Accordingly, we have extensively revised the manuscript in line with comments to clarify the overall message to highlight our key points. In addition, we have reviewed the figures and figure legends to provide further experimental information and detail.

REVIEWERS' COMMENTS:

Reviewer #3 (Remarks to the Author):

Minor

1. Figure 1 description in the Results section should include some of the observations in more detail. For example, (a) some of the GWAS genes should be listed- after careful consideration of those that may be most meaningful to discuss - in the text and commented on which cell subsets expressed them. (b) after CAREFUL CONSIDERATION of most meaningful observations something like: the pericytes were marked by X, Y and Z genes, were as fibroblasts by The single-cell RNAseq and CyTOF data aligned with THESE markers for THESE cell types, where CyTOF further elucidated cell surface expression of THESE markers for THIS CELL SUBSET.

2. Consider putting into a main figure the fibroblast subset FUNCTIONAL data via Thp-1 chemotaxis data.

Layton et al (2020) Reply to reviewer's comments

Reviewer #3 (Remarks to the Author):

Minor

1. Figure 1 description in the Results section should include some of the observations in more detail. For example, (a) some of the GWAS genes should be listed- after careful consideration of those that may be most meaningful to discuss - in the text and commented on which cell subsets expressed them. (b) after CAREFUL CONSIDERATION of most meaningful observations something like: the pericytes were marked by X, Y and Z genes, were as fibroblasts by The single-cell RNAseq and CyTOF data aligned with THESE markers for THESE cell types, where CyTOF further elucidated cell surface expression of THESE markers for THIS CELL SUBSET.

Many thanks for this comment. We agree this point required further clarification and we have updated our discussion of these results as follows: Marker genes for pericytes included *JAG1* and *MCAM*, *SFPR4* and *PLA2G2A* for fibroblasts, and myofibroblasts were marked by *MMP14* and *MAFB*. The single cell RNA-seq and CyTOF aligned with significant markers for stromal cell types, where CyTOF further elucidated cell surface expression of these markers for fibroblasts, myofibroblasts and pericytes.

2. Consider putting into a main figure the fibroblast subset FUNCTIONAL data via Thp-1 chemotaxis data.

Many thanks. The THP-1 chemotaxis data have now been included in the main figure 2.